# A Brief Review of Bolus Osmotherapy Use for Managing Severe Traumatic Brain Injuries in the Pre-Hospital and Emergency Department Settings

Vignesh Raman [1,*], Matthew Bright [2] and Gary Mitchell [3]

1   Department of Neurosurgery, Royal Brisbane and Women's Hospital, Herston, QLD 4029, Australia
2   Intensive Care Services, Royal Brisbane and Women's Hospital, Herston, QLD 4029, Australia; matthew.bright@health.qld.gov.au
3   Emergency and Trauma Centre, Royal Brisbane and Women's Hospital, Herston, QLD 4029, Australia; gary.mitchell2@health.qld.gov.au
*   Correspondence: vignesh.raman@health.qld.gov.au

**Abstract:** Background: Severe traumatic brain injury (TBI) management begins in the pre-hospital setting, but clinicians are left with limited options for stabilisation during retrieval due to time and space constraints, as well as a lack of access to monitoring equipment. Bolus osmotherapy with hypertonic substances is commonly utilised as a temporising measure for life-threatening brain herniation, but much contention persists around its use, largely stemming from a limited evidence base. Method: The authors conducted a brief review of hypertonic substance use in patients with TBI, with a particular focus on studies involving the pre-hospital and emergency department (ED) settings. We aimed to report pragmatic information useful for clinicians involved in the early management of this patient group. Results: We reviewed the literature around the pharmacology of bolus osmotherapy, commercially available agents, potential pitfalls, supporting evidence and guideline recommendations. We further reviewed what the ideal agent is, when it should be administered, dosing and treatment endpoints and/or whether it confers meaningful long-term outcome benefits. Conclusions: There is a limited evidence-based argument in support of the implementation of bolus osmotherapy in the pre-hospital or ED settings for patients who sustain a TBI. However, decades' worth of positive clinician experiences with osmotherapy for TBI will likely continue to drive its on-going use. Choices regarding osmotherapy will likely continue to be led by local policies, individual patient characteristics and clinician preferences.

**Keywords:** osmotherapy; hypertonic saline; mannitol; traumatic brain injury; intracranial pressure; pre-hospital medicine

## 1. Introduction

Severe traumatic brain injuries (TBIs) are the leading cause of trauma-related deaths worldwide, but in recent years there has been an observed reduction in mortality rates [1,2]. Owing to a significant portion of TBI survivors living burdened with major physical, cognitive and psychological morbidities, there has been a paradigm shift in TBI research, with a focus on interventions for improving both mortality outcomes and functional recovery [3]. Given the time-sensitive nature of definitive treatment and prognosis in trauma, it can be appreciated that these interventions may often need to occur as early as the pre-hospital and emergency department (ED) settings [4,5].

Osmotherapy has been used as an intervention for temporarily reducing elevated intracranial pressure (ICP) in the context of TBI [1]. Introduced over 100 years ago, the role of osmotherapy in the pre-hospital and ED settings in the management of TBI still remains a topic of contention [2]. Although widely used in the pre-hospital and ED settings, neither mannitol nor hypertonic saline (HTS) have been supported by randomised controlled trials

(RCTs) as providing long-term benefits amongst patients with TBI [4]. Further, the debate around the superiority of HTS compared to mannitol in TBI has been on-going for over 20 years, and recent studies lean towards a benefit with HTS, albeit weakly supported [1]. There has also been some interest shown in recent years in opting for alternative hypertonic sodium-containing solutions, such as sodium bicarbonate (SB) [1,6,7].

The purpose of this brief review is to summarise pragmatic information related to the use of osmotherapy for TBI in the pre-hospital and ED settings. The authors highlight osmotherapy pharmacology, commercially available agents, indications and dosing, as well as summarising the current evidence base and established guidelines.

## 2. Pharmacology, Agents and Dosing Regimens

Intravenous osmotherapy works on the principle that, following administration of an agent with significantly higher osmolality than native plasma (typically 275–295 mOsm/L), overall plasma osmolality rapidly increases and fluid shifts occur from the extravascular (both intracellular and interstitial) spaces into the intravascular space [7]. In the case of TBI, osmotherapeutic agents reduce ICP by essentially dehydrating the neurons and endothelial cells in regions with an intact blood–brain barrier [7]. Neuronal dehydration leads to reduced cerebral swelling, whilst endothelial dehydration leads to thinner capillary walls and an improved diffusion gradient for oxygen uptake by the brain [7]. The movement of water into the intravascular space also reduces blood viscosity, further facilitating improved cerebral blood flow [7]. The three most widely used osmotherapeutic agents that reduce ICP through this mechanism are mannitol, HTS and SB (see Table 1).

**Table 1.** Comparison of commonly used osmotherapy agents.

| Agent | Mannitol | Hypertonic Saline | Sodium Bicarbonate |
|---|---|---|---|
| Concentrations and Dosing | 10%—0.25–2 g/kg over 30–60 min <br> 20%—20–80 mL | 3%—2 mL/kg over 10–15 min <br> 7.5%—1–2 mL/kg <br> 23.4%—20 mL | 8.4%—1–2 ampoules (50–100 mL) over 10 min |
| Mechanisms of Action | • Osmotic diuretic <br> • Suppresses arginine vasopressin release <br> • Promotes atrial natriuretic peptide release <br> • Free radical scavenger | • Hyperosmolar substance promoting shift of intravascular water from other compartments (e.g., brain tissue) | • Hyperosmolar substance promoting shift of intravascular water from other compartments (e.g., brain tissue) |
| Advantages | • No need for central venous access <br> • Free radical scavenger function | • Rapid onset <br> • No rebound oedema <br> • Response up to 12 h <br> • Some immunomodulatory effects | • No need for central venous access <br> • Does not cause metabolic acidosis |
| Disadvantages | • Systemic hypotension <br> • Acute kidney injury at high doses <br> • Rebound oedema | • Hypernatraemia <br> • Hyperchloraemic metabolic acidosis <br> • Hyperoncotic haemolysis <br> • Need for central venous access <br> • Risk of acute pulmonary oedema | • Hypernatraemia |

Mannitol is a naturally occurring carbohydrate isomer of sorbitol that is extracted from the secretions of the flowering ash tree [7,8]. It is a large molecule that weighs 182 daltons and undergoes free glomerular filtration without biotransformation. Commercially, it is available as either a 10% (1 g/mL, 549 mOsm/L) or 20% (2 g/mL, 1098 mOsm/L)

preparation [8]. A unique mechanism for further neuroprotection by mannitol in TBI is its capacity to scavenge free radicals that lead to secondary brain injury [7]. However, mannitol functions as an osmotic diuretic that acts at nephrons by creating an osmotic pressure gradient at the proximal convoluted tubule and the descending loop of Henle to minimise water reabsorption and produce a dilute tubular sodium concentration [7]. Additionally, mannitol suppresses the release of endogenous arginine vasopressin and promotes the release of atrial natriuretic peptide, leading to the production of a high volume of dilute urine [7]. Mannitol has somewhat fallen out of favour due to its theoretical potential to cause hypotension, a well-established negative prognostic indicator in severe TBI [9], but bolus dosing mannitol in the pre-hospital setting has not been observed to alter blood pressure [10].

HTS refers to any preparation of sodium chloride (NaCl) solution that exceeds physiological concentration—readily available HTS concentrations can range between anywhere from 3% to 30% (1027–10,000 mOsm/L) [7]. Dextrans are also often added in some HTS formulations as plasma volume expanders—one study found that dextrans also modulated inflammatory and coagulation responses to TBI and reduced secondary brain injury [11]. An additional mechanism of neuroprotection provided by HTS is its ability to increase blood pressure by increasing intravascular volume, which leads to increased cerebral perfusion pressure and cerebral blood flow [2]. HTS also has a pharmacokinetic benefit over mannitol, with rapid onset of action within five minutes and reported effects for up to twelve hours in certain patients [2]. Interestingly, however, although HTS has been shown to improve cerebral perfusion, it has not been shown to improve brain tissue oxygenation [12].

SB, an alternate hypertonic sodium-containing solution, is a less popular option for bolus osmotherapy in TBI [6]. Given that the osmolality of 8.4% SB is 2000 mOsm/L, which is equivalent to approximately 5.8% of NaCl, SB can be considered as almost twice as potent as 3% HTS [6]. Thus, one ampoule (50 mL) of 8.4% SB is approximately the same as bolus dosing 100 mL of 3% HTS. Another argument for SB over HTS is that it does not lead to hyperchloraemic metabolic acidosis, particularly with repeat bolus dosing [6]. The clinical significance of hyperchloraemic metabolic acidosis is unclear, but given that TBI patients are already at higher risk of acidosis, particularly in instances of multi-trauma, SB may be a good alternative to HTS [13].

Hypertonic sodium lactate has also been explored for osmotherapy as either an infusion or bolus for elevated ICP management in TBI but, again, partly due to concerns of worsening acidosis, it has not gained much popularity [14,15].

## 3. Indications, Response Targets and Adverse Effects

The main use for osmotherapy in TBI is temporarily managing elevated ICP prior to definitive treatment either via surgical decompression or barbiturate coma with multimodal monitoring in the ICU (BTF guidelines) [7]. Elevated ICP is clinically inferred from signs of transtentorial herniation with or without brainstem compression, such as new mydriasis, hypertension with irregular respirations and reflex bradycardia (Cushing's response) and/or new motor posturing [1]. Resuscitation amongst hypotensive patients with TBI is also considered an indication for HTS or SB, as it can promote fluid shifts into the intravascular space and improve cardiac preload [1,16]. Adequate resuscitation in the pre-hospital setting has profound importance given that a single episode of systolic blood pressure < 90 mmHg is associated with double the likelihood of mortality in TBI [9]. Using osmotherapy prophylactically in patients with severe TBIs and multi-modal monitoring has been explored in the intensive care unit (ICU) setting, but it has not been similarly explored in the pre-hospital or ED settings [8].

With regard to response targets and monitoring in the pre-hospital and ED settings, clinicians are limited to serial examinations of pupils and haemodynamic changes with manual blood pressure and electrocardiographic monitoring post-bolus dosing [17]. Improved reactivity of sluggish pupils or changes in mean arterial pressure can be considered responses, but whether such responses are adequate and the appropriate minimal timing

interval between doses remain unclear [16,17]. Furthermore, reports regarding the duration of response to bolus osmotherapy are highly variable between patients [17]. There are studies that suggest that when a response to HTS lasts longer than two hours, it is associated with decreased mortality and improved functional outcomes [17].

Mannitol was traditionally the first-line osmotherapy option in patients with TBI, but concerns about hypotension, which usually occurs approximately 45 min after bolus dosing due to osmotic diuresis, have led to concerns regarding the risk of secondary brain injury [7]. There are also some reports of repeated dosing of mannitol leading to its passage from serum to brain tissue via the damaged blood–brain barrier, causing rebound oedema and increased ICP [18]. Further, particularly at higher doses (usually exceeding 200 g in 24 h), mannitol can also lead to acute kidney injury secondary to intravascular volume depletion and intrarenal vasoconstriction, as well as causing contraction alkalosis, hypochloraemia, hypokalaemia and hypomagnesemia [7].

In contrast, HTS and SB cause intravascular volume expansion, which can lead to acute pulmonary oedema [7]. Intensivists advocate for concurrent administration of intravenous furosemide to promote concurrent diuresis in the ICU setting, but such a practice has not been universally accepted in the pre-hospital or ED settings due to concerns regarding profound hypotension [19]. Furthermore, HTS has been associated with rates of acute kidney injury higher than those resulting from 0.9% NaCl when given at equal doses [2]. HTS can also lead to hyperchloremic metabolic acidosis, potentiating further bleeding in trauma, and hypokalaemia, whilst SB can lead to metabolic alkalosis [2,6]. It also remains unclear what the maximum tolerable serum sodium value and osmolality are prior to causing clinically relevant dehydration or acute kidney injury—the literature recommends not exceeding a serum sodium value of 150 mmol/L and serum osmolality of 320 mOsm/L, but both these values have been challenged and somewhat safely exceeded before [8].

Consideration of the patient's blood pressure and intravascular volume status, particularly in the context of polytrauma, can guide the choice between a diuretic, such as mannitol, or volume expander, such as HTS or SB, for the management of elevated ICP in TBI. Other invaluable information for deciding the appropriate agent includes cardiac function and renal function but, understandably, access to such information is not always available in the pre-hospital and ED settings.

## 4. Evidence Base for the Pre-Hospital and ED Settings

The debate around the superiority of HTS compared to mannitol has been on-going for over 20 years and remains without a definitive conclusion despite several systematic reviews, including two Cochrane reviews and five meta-analyses [18,20–24]. The current impression is that, in the context of TBI, there is no single clearly superior agent. An RCT in the UK entitled "Sugar or Salt" comparing the use of bolus HTS and mannitol for the management of elevated ICP in the ICU setting is currently underway [25]. However, there are no similar RCTs focused on the pre-hospital and ED settings.

Cook et al. recently conducted an expert panel review of the literature evaluating the use of HTS and mannitol in cases of pre-hospital management of TBI, with a particular focus on ICP reduction, the impact on neurological outcomes and treatment of hypotension [1]. With regard to ICP reduction, some meta-analyses comparing HTS and mannitol have found no difference between the two agents and others favour HTS over mannitol [26]. However, evidence from direct comparisons and cross-over and rescue therapy RCTs support HTS over mannitol [26]. Several studies have compared osmotherapy agents and the impact on neurological outcomes and found no benefits [20,27–29]. Other studies have investigated the mortality and/or morbidity benefits of mannitol and HTS employed in the pre-hospital setting for resuscitation and found no benefit [10,16,30].

In summary, although mannitol, HTS and SB can address the physiological abnormalities associated with elevated ICP and cerebral oedema, there is only weak evidence that employing these agents in either pre-hospital or hospital settings for TBI improves mortality or long-term functional outcomes amongst survivors. There is also weak evidence

to support the prophylactic use of osmotherapy for patients without clinical features of transtentorial herniation or hypotension.

## 5. Guideline Recommendations for the Pre-Hospital and ED Settings

Summarised in Table 2, there are currently no guideline recommendations specifically supporting the use of osmotherapy in the pre-hospital or ED settings when managing patients with TBI. There are guidelines, primarily in the context of ICU setting-based data, published by the Neurocritical Care Society (NCS), who updated their recommendations in 2020 and suggested using HTS over mannitol for the initial management of elevated ICP amongst patients with TBI [1]. However, the NCS did acknowledge that this was a conditional recommendation supported by low-quality evidence. Regarding the pre-hospital setting, the NCS advised against using HTS or mannitol for the purpose of improving neurological outcomes, but this was based on moderate- and low-quality evidence, respectively [1].

**Table 2.** Summary of guideline recommendations for osmotherapy in the pre-hospital and ED settings.

| Guidelines | Country | Publication Year | Recommendation |
|---|---|---|---|
| Neurocritical Care Society | USA | 2020 | • HTS and mannitol should not be given in the pre-hospital setting specifically for improving neurological outcomes |
| Brain Trauma Foundation | USA | 2017 | • No recommendations regarding osmotherapy in the pre-hospital or ED settings |
| Head Injury, the Early Management | UK | 2014 | • No recommendations regarding osmotherapy in the pre-hospital or ED settings |
| Trauma Audit and Research Network | UK | 2022 | • No recommendations regarding osmotherapy in the pre-hospital or ED settings |

The Brain Trauma Foundation (BTF) guidelines are widely considered as the gold standard for the design and development of local TBI management protocols [30]. In the most recent fourth edition of the BTF guidelines, published in 2017, the authors undertook a systematic review of studies comparing various osmotherapy agents and conceded that there was insufficient evidence to support the clinical benefits of osmotherapy or to recommend the superiority of any one agent over another [30]. Furthermore, in 2017, the authors of the BTF guidelines removed the previously published recommendation of using mannitol for patients without ICP monitors but with signs or symptoms of progressive neurological decline or transtentorial herniation due to weak standards of evidence [30].

Interestingly, in both the "Head Injury, the Early Management" study, published by the National Institute for Health and Care Excellence, and the "Trauma Audit and Research Network" study altogether omit any reference to osmotherapy for acute management of TBI [31].

## 6. Conclusions

In conclusion, there is no strong evidence-based argument for the implementation of bolus osmotherapy in the pre-hospital or ED settings for patients who sustain a TBI. Additionally, if osmotherapy is to be implemented, there are no definitive answers concerning which osmotic agent is the most ideal and which has the strongest evidence base. Other unanswered questions concern how frequently a patient can be re-dosed, whether osmotic agents can be safely interchanged for the same patient and when clinicians should stop

osmotherapy. Given that there is unlikely to be adequate equipoise to experiment with whether or not osmotherapy should be used amongst patients with TBI, decades' worth of clinician experience with osmotherapy in TBI will likely continue to drive its on-going use. Furthermore, debates regarding the choice of osmotherapy will continue to be led by local policies, individual patient characteristics and clinician preferences.

**Author Contributions:** Conceptualization, V.R., M.B. and G.M. methodology, V.R.; writing—original draft preparation, V.R.; writing—review and editing, V.R., M.B. and G.M.; project administration, V.R. All authors have read and agreed to the published version of the manuscript.

**Funding:** This research received no external funding.

**Institutional Review Board Statement:** Not applicable.

**Informed Consent Statement:** Not applicable.

**Data Availability Statement:** Not applicable.

**Conflicts of Interest:** The authors declare no conflict of interest.

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
