# Peer review of "A Brief Review of Bolus Osmotherapy Use for Managing Severe Traumatic Brain Injuries in the Pre-Hospital and Emergency Department Settings"

_traumacare, doi:10.3390/traumacare2030035_

Round 1

Reviewer 1 Report

Thank you for allowing me to review this good piece of brief review contributing to the knowledge about using bolus osmotherapy for managing severe traumatic brain injuries. The manuscript generally looks alright

- the introduction and results sections are clear and described in sufficient detail to understand the topic discussed in brief. The conclusions are a reasonable extension of the result and have pointed out the unanswered questions and future directions. A bit ambiguous that this work was defined as “a narrative review” in the abstract section though, given that narrative review often considers something different from brief review. Overall I suggested this paper being accepted after minor revision.

Author Response

  • Thank you kindly for taking our manuscript into consideration and your positive comments.
  • We have changed ‘narrative’ review to ‘brief’ review in the abstract to avoid confusion, as the definition of a brief review is more in keeping with this body of work.

Reviewer 2 Report

- Please add town and country to your affiliations

- Page 1, line 44 omit "not" (or revise the sentence)

Author Response

  • Thank you kindly for taking our manuscript into consideration.
  • We have included our town and country to affiliations- our institution is in Brisbane, Australia.
  • We have completely re-written the sentence referred to so that we could better communicate our message to the readers.

Reviewer 3 Report

It is a very well written review paper. The only my concern is that the number of references for a review paper is relatively small and perhaps should be increased. Secondly, it will be interesting if authors provide some more information regarding some other therapies tested in clinical trials (if any are). it will be very interesting for a common reader to compare those results/data. Otherwise it is an interesting paper and will be of particular interest to ED physicians. 

Author Response

  • Thank you for reviewing our manuscript and we agree, our aim was to provide an interesting read for ED physicians.
  • We have addressed the recommendation of increased references by including a further 5 references to this brief literature review.
  • Other therapies tested in clinical trials have been hypertonic sodium lactate, as both an infusion and bolus dosing to manage raised ICP- we have now included this in the manuscript. There is also an ongoing trial titled ‘Salt or Sweet’, comparing HTS with mannitol in the ICU severe TBI population- unfortunately this trial has not been conducted in the pre-hospital or emergency department settings.

Reviewer 4 Report

Reviewer comments:

Comments to the Author

This review by Dr. Vignesh Raman et al., highlighted the importance of severe traumatic brain injury (TBI) management. Here, they tried to summaries pragmatic information related to the use of osmotherapy for TBI in the pre-hospital and emergency department settings, including pharmacology, indications, commercially available agents, potential pitfalls, supporting evidence and established guidelines.

·       This article is written very well, and the incidences were provided clearly. This review greatly emphasizes on the clinician experiences with osmotherapy for TBI.

Minor comment:

·       Can the authors provide a brief summary of the most highlighted facts of this review at the end of the introduction?

·       Authors are advised to go through the proper spell check, grammatical errors and typographical errors.

Author Response

  • As recommended, we have included in our introduction that the important facts we uncovered were:
    • There is limited evidence for bolus osmotherapy in the pre-hospital/ED setting.
    • There is ongoing contention over the superior osmotherapy agent.
    • There is ongoing contention about dosing and re-dosing of bolus osmotherapy.
  • Thank you kindly for your positive feedback. We have gone through and checked the manuscript for spelling, grammar and typographical errors as recommended.